# Robust Selective Activation with Randomized Temporal K-Winner-Take-All in Spiking Neural Networks for Continual Learning

**Jiangrong Shen**[1,2]   **Liang Zhao**[2,3]   **Qi Xu**[4]   **Yuqi Yang**[5]   **Liangjun Chen**[2,7]
**Gang Pan**[6]   **Badong Chen**[2,7*]

[1]School of Computer Science and Technology, Xi'an Jiaotong University
[2]National Key Laboratory of Human-Machine Hybrid Augmented Intelligence, Xi'an Jiaotong University
[3]College of Artificial Intelligence, Xi'an Jiaotong University
[4]School of Computer Science and Technology, Dalian University of Technology
[5]School of Cyberspace Security and Computer Science, Hebei University
[6]State Key Lab of Brain-Machine Intelligence, Zhejiang University
[7]Institute of Artificial Intelligence and Robotics, Xi'an Jiaotong University

## Abstract

The human brain exhibits remarkable efficiency in processing sequential information, a capability deeply rooted in the temporal selectivity and stochastic competition of neuronal activation. Current continual learning in spiking neural networks (SNNs) faces a critical challenge: balancing task-specific selectivity with adaptive resource allocation and enhancing the robustness with perturbations to mitigate catastrophic forgetting. Considering the intrinsic temporal dynamics of spiking neurons instead of traditional K-winner-take-all (K-WTA) based on firing rate, we explore how to leave networks robust to temporal perturbations in SNNs on lifelong learning tasks. In this paper, we propose Randomized Temporal K-winner-take-all (RTK-WTA) SNNs for lifelong learning, a biologically grounded approach that integrates trace-dependent neuronal activation with probabilistic top-k selection. By dynamically prioritizing neurons based on their spatiotemporal relevance, RTK-WTA SNNs emulate the brain's ability to modulate neural resources in spatial and temporal dimensions while introducing controlled randomness to prevent overlapping task representations. The proposed RTK-WTA SNNs enhance inter-class margins and robustness through expanded feature space utilization theoretically. The experimental results show that RTK-WTA surpasses deterministic K-WTA by 3.07–5.0% accuracy on splitMNIST and splitCIFAR100 with elastic weight consolidation. Controlled stochasticity balances temporal coherence and adaptability, offering a scalable framework for lifelong learning in neuromorphic systems.

## 1 Introduction

The human brain exhibits remarkable efficiency in processing sequential information, a capability deeply rooted in the temporal dynamics of neuronal activation Kandel & Hawkins (1992); Bassett et al. (2011). Biological neurons operate under a principle of selective activation, where sparse subsets of cells fire in temporally coordinated patterns to encode specific stimuli or tasks Kudithipudi et al. (2022). This sequential, task-dependent recruitment of neuronal populations is a hallmark of continual learning in biological systems, enabling humans to acquire new skills over time while preserving previously learned knowledge. The selectivity is not static but evolves dynamically across time scales, governed by mechanisms such as short-term synaptic plasticity and adaptive K-winner-take-all (K-WTA) competition. Crucially, biological WTA circuits integrate both deterministic competition, where neurons with stronger inputs dominate, and controlled stochasticity, ensuring that activation patterns remain flexible and robust against overlapping task demands. This interplay

---

*Corresponding author: chenbd@mail.xjtu.edu.cn.

between temporal precision and randomness allows the brain to avoid rigid specialization, instead maintaining a balance between task-specific efficiency and generalization across contexts.

In artificial intelligence, continual learning aims to emulate the biological ability to acquire new knowledge over time without sacrificing performance on previously learned tasks. However, unlike conventional training paradigms that assume access to all data simultaneously, continual learning must operate on sequential and often non-overlapping task streams. This setting makes neural networks highly susceptible to catastrophic forgetting, where optimizing for a new task overwrites crucial representations established for earlier tasks. Unlike traditional artificial neural networks (ANNs), SNNs operate through discrete spikes—representing the firing or non-firing of spiking neurons at specific time points—which leads to richer and more complex spatiotemporal dynamics Maass (1997); Subbulakshmi Radhakrishnan et al. (2021); Bu et al. (2023). This event-driven processing enables SNNs to capture temporal information more effectively, maintain memory more robustly, mimic the behavior of biological neural systems more closely, and support energy-efficient deployment Zhao et al. (2025); Bu et al. (2025); Xu et al. (2025). These biological and computational properties make SNNs a particularly promising substrate for continual learning, since the spatiotemporal dynamics could provide naturally separated, temporally distinct activation patterns that reduce representational overlap and mitigate interference across sequential tasks.

Continual learning in SNNs represents a paradigm shift in artificial intelligence by emulating the human brain's ability to acquire and retain knowledge over time while adapting to new information without catastrophic forgetting. To this end, biologically-inspired strategies such as synaptic plasticity Kaiser et al. (2020); Schmidgall et al. (2021), neural reorganization Han et al. (2025), weight consolidation Golden et al. (2022); Ning et al. (2023), and dynamic routing Putra & Shafique (2021) have been explored to optimize continual learning. These approaches leverage SNNs' inherent biological and computational properties such as spatiotemporal dynamics. However, most of these models ignore the robustness of SNNs during spatiotemporal activation. The random localized competition reduces representational overlap across tasks and has the potential to improve resistance to interference, thereby enhancing functional robustness of SNNs in dynamic environments. Likewise inspired by the biological robustness mechanisms, we further explore how to build robust SNNs capable of continual learning by leveraging temporal selectivity and stochastic K-WTA dynamics. Most of the traditional K-WTA implementations in SNNs adopt deterministic spatial competition, selecting neurons with the highest instantaneous firing rates or membrane potentials. Those approaches neglect the temporal richness of spiking signals and fail to emulate the brain's ability to modulate activation patterns across time Shen et al. (2023). Recent SNN studies have also highlighted that explicitly modeling time-varying spatiotemporal dynamics is crucial for effective information integration Shen et al. (2025); Xu et al.. By exploiting the temporal trace property and designing the trace-based K-WTA, Shen et al. (2024) have demonstrated the more abundant representation ability of SNNs with the addition of temporal competition. However, the robustness of that model is limited since the temporal trace selections are deterministic. The biological neurons rely on temporal traces of past activity to guide future responses, enabling them to prioritize inputs that align with ongoing temporal contexts. Integrating these traces with random top-k selection, where a stochastic subset of neurons is activated, could mirror the brain's adaptive K-WTA mechanisms. By coupling temporal trace accumulation with controlled randomness, SNNs could dynamically allocate neural resources to tasks while mitigating catastrophic interference caused by overlapping activations McCloskey & Cohen (1989); French (1999).

Recent advances in SNNs have yet to fully exploit this synergy. Most existing selective activation strategies focus on spatial sparsity or static firing thresholds, overlooking the temporal dynamics. Deterministic K-WTA rules, while effective for isolated tasks, risk rigid binding of neurons to specific patterns, leaving networks vulnerable to input perturbations and catastrophic forgetting when tasks share similar characteristics Lynch et al. (2019). In contrast, a trace-based random top-k mechanism introduces two critical innovations: (1) temporal random selectivity, where neurons are prioritized based on their accumulated activity traces, ensuring alignment with the sequential structure of tasks, and (2) stochastic competition, which injects diversity into activation patterns to prevent task representations from converging to overlapping subspaces. This biologically grounded approach not only enhances robustness but also provides a scalable framework for lifelong learning in SNNs, addressing a fundamental gap in current neuromorphic architectures.

We argue that this issue arises because the architecture does not recognize the sequential nature of learning. Existing SNN models often treat tasks as discrete, independent events, employing activation

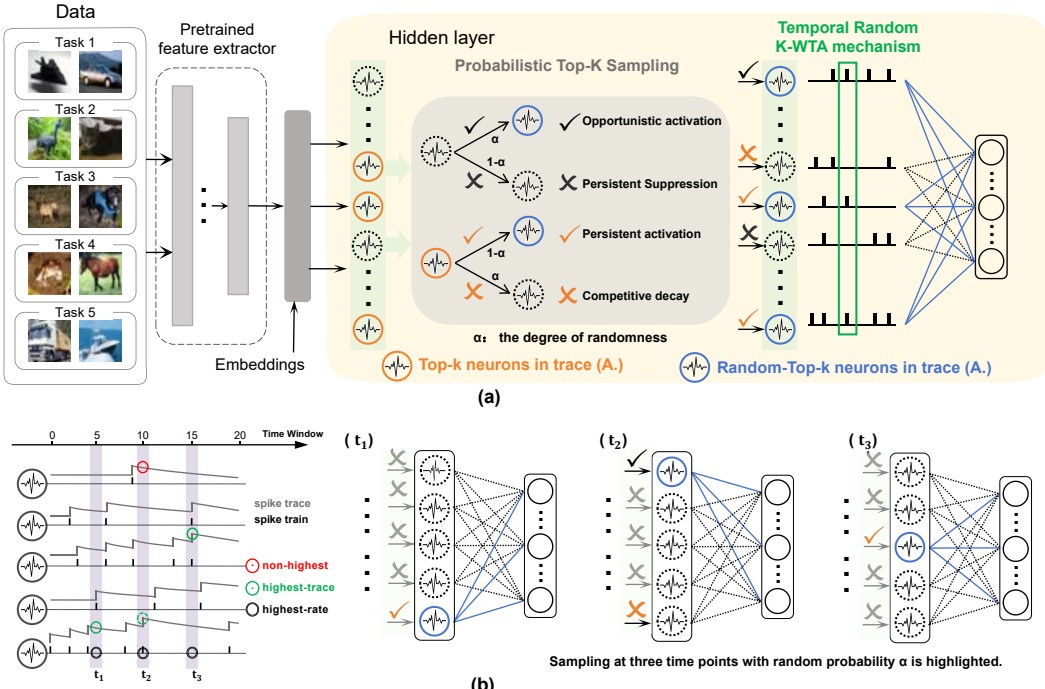

Figure 1: (a) The architecture of the proposed randomized temporal winner-take-all (RTK-WTA) for SNNs. (b) An example to show the working mechanism of the RTK-WTA mechanism for SNNs.

strategies optimized for static inputs rather than evolving temporal contexts. This oversight leads to two critical failures. First, deterministic K-WTA mechanisms lack the adaptability to dynamically reweight neurons as task sequences unfold, forcing networks to overwrite previously learned representations. Second, without explicit decoupling of task-specific neural pathways—enabled by trace-modulated stochasticity—even sparse activations may interfere across tasks, accelerating catastrophic forgetting. Our work bridges this gap by unifying temporal trace dynamics with random top-k selection, enabling SNNs to emulate the brain's ability to balance specificity and flexibility in sequential learning. The contributions are summarized as follows:

- We propose a biologically inspired randomized temporal K-winner-take-all (RTK-WTA) for SNNs in continual learning scenarios. RTK-WTA integrates temporally accumulated neuronal traces with probabilistic top-k to realize robust selective activation. This design dynamically prioritizes neurons based on temporal relevance while introducing controlled stochasticity to reduce overlapping activations and mitigate catastrophic forgetting.

- Through theoretical and empirical analysis, RTK-WTA enlarges the effective spatiotemporal feature space of SNNs. By aligning neural activation with task-specific temporal dynamics, the method enhances the diversity of internal representations and facilitates the separation of overlapping task features, thus promoting stable knowledge retention across tasks.

- Experiments on continual learning benchmarks—including SplitMNIST, SplitCIFAR-10, and SplitCIFAR-100 datasets demonstrate that our model consistently outperforms deterministic Top-K baselines by margins of up to 10.05%. Furthermore, RTK-WTA exhibits strong resilience to input perturbations and task interference by explicitly decoupling neural pathways through trace-driven stochastic activation, enabling more robust and scalable continual learning.

## 2 METHODS

Figure 1(a) illustrates the architecture of the RTK-WTA in SNNs, which employs trace-based probabilistic top-k selection according to neuronal activation dynamics. This mechanism enhances

the model's generalization capacity through expanded feature space utilization while maintaining alignment with selective activation mechanisms observed in biological neural systems.

Taking three temporal instances of neuronal activation as an example, our RTk-WTA mechanism selects three distinct neuron types as illustrated in Figure 1 (b): highest-rate, highest-trace, and non-highest neurons. Combined with the spiking firing of five neurons in the left part of Figure 1 (b), the rate-based K-WTA solely activates one neuron ($5_{th}$ neuron) across three different time points ($t_1, t_2, t_3$). While the only trace-based SNNs without random K-WTA activate two neurons ($3_{rd}$ and $5_{th}$ neurons) across these time points. Our RTK-WTA SNNs activate three neurons ($1_{st}, 3_{rd}$ and $5_{th}$ neurons) across these time points. This demonstrates our mechanism's capacity to promote firing across neurons with varying states, thereby enabling exploration of a broader feature space. Therefore, for continual learning scenarios, the activation diversity in our RTK-WTA SNNs maintains redundant neuronal activation patterns, forming complementary memory representations during sequential task learning. When new tasks interfere with partial neurons, the remaining neuronal populations preserve historical task information, thus mitigating catastrophic forgetting.

## 2.1 RANDOMIZED TEMPORAL TRACE-BASED K-WTA SELECTION

In this paper, we regard the trace as the indicator for random temporal K-WTA. The trace in spiking neurons provides an efficient way to online estimate the mean firing rate related to the spike train Morrison et al. (2008), which bridges the gap between time scale and action potential in the plasticity theory. The trace is updated with each spike-firing behavior and decays between spike-firing behaviors, which is a more concise intrinsic information expression for spiking neurons compared to the pure spiking firing rates. Therefore, the accumulation manner of trace provides a stable metric for stochastic selection, effectively mitigating task interference without sacrificing temporal resolution.

We begin by clearly defining the dynamics of the neuronal trace $tr_i[t]$ in the spiking neuron model at discrete time step $t$:

$$tr_i[t+1] = tr_i[t] - \frac{tr_i[t]}{\tau} + S_i[t+1],\tag{1}$$

where $\tau$ is the decay constant of the neuronal trace and $S_i[t+1] \in 0, 1$ denotes the generated spike train at time $t+1$. This trace provides a temporally integrated representation of neuronal spiking activity, offering robust and distinctive temporal features necessary for efficient continual learning.

Based on that, we propose the randomized trace-based Top-K (RTK) selection based on the neuronal traces at each time step in order to enhance the robustness and improve the generalization capability of SNNs for continual learning tasks. For a network of $d$ neurons, the RTK selection generates a binary mask:

$$Mask[t] = RTK(tr[t]),\tag{2}$$

where the probability of selecting neuron $i$ at time $t$ is defined as

$$P(RTK(tr_i[t]) = 1) = \begin{cases} (1-\alpha)/K, & if\ tr_i[t]\ is\ top\ K\ trace; \\ \alpha/(d-K), & otherwise. \end{cases}\tag{3}$$

where $\alpha \in [0, 1]$ controls randomness during the trace-based top K selection and $K$ is the number of neurons selected at each time step.

The above randomized trace-based K-WTA provides a larger feature space than traditional K-WTA by extending the top K selection from spatial dimension to spatiotemporal dimension and, meanwhile, bringing suitable randomness to explore a larger manifold on neuron selection and avoid dropping into a local minimum.

Then the spiking output after applying RTK-WTA is

$$S^*[t] = S[t] \cdot Mask[t],$$

In continual learning settings, this selective and stochastic activation across different time steps enables differentiated neural responses for sequential tasks, mitigating overlap in active subspaces

and reducing interference. Thus, by combining with the K-WTA neuron selection in spatiotemporal, it creates implicit task separation in the temporal domain without explicit task labels.

## 2.2 GENERALIZATION ANALYSIS OF RTK-WTA SNNs

The recursive update rule in equation 1 captures the short-term memory of spiking events: each spike adds to the trace, while the trace decays over time if no spikes occur. Unlike conventional neurons that reset their state completely between time steps, the trace persistence allows the network to maintain short-term memory of recent spikes while gradually forgetting older events. It implicitly performs an exponential moving average over the spike history. The decay constant $\tau$ controls the effective memory window - smaller values cause faster forgetting for rapid pattern detection, while larger values enable longer temporal integration. To make this integration explicit, we consider its unrolled form over a temporal window of length $T$. This leads to the definition of an integrated trace vector:

$$Tr_i^{(T)} = \sum_{t=1}^{T}(1 - \frac{1}{\tau})^{T-t}S_i[t].$$

(4)

This representation accumulates past spikes with an exponentially decaying weight, emphasizing recent activity. Recent spikes (when t approaches T) contribute more significantly to the current trace value than distant spikes. The weighting scheme creates a continuous spectrum of time sensitivity - the same spike pattern occurring at different temporal positions will produce distinguishable trace states, enabling position-invariant pattern recognition. Compared to the recursive trace update, this formulation provides a static, interpretable form of the trace for use in theoretical analysis, such as feature space volume and margin estimation.

We now analyze the generalization enhancement by considering the temporal integration of traces. The overall temporal feature representation for all neurons is $Tr^{(T)} = (Tr_1^{(T)}, ..., Tr_d^{(T)})$. Due to the RTK-WTA mechanism, the effective number of possible activation patterns over the temporal window expands significantly.

At each time step $t$, the effective number of neuron activation combinations, i.e., the possible configurations of $K$ neurons being selected out of $d$ with stochastic probabilities are approximated by

$$\mathcal{N}_t = \binom{d}{K}\left[(1 - \alpha) + \frac{\alpha K}{d - K}\right]^K.$$

(5)

Here $\binom{d}{K}$ is the total number of possible $K$-sized subsets from $d$ neurons (structural degrees of freedom), and the term in brackets approximates the expected selection probability per neuron when considering both top-K and non-top-K candidates under randomness $\alpha$.

In deterministic Top-K, only a small subset of the $\binom{d}{K}$ configurations are reachable due to consistent selection bias toward a fixed group of neurons. However, RTK-WTA introduces variability across time, allowing a significantly larger set of combinations to be explored. This probabilistic diversification reflects the spatiotemporal nature of SNNs—where not only spatial configurations are considered by firing rate that integrates temporal information across all time steps, but also dig into the spatiotemporal dynamics by selectively activating neurons in each time step in a random way to extend the representation space.

When this mechanism is accumulated over a temporal window of $T$ steps, the spatiotemporal feature space becomes exponentially rich:

$$\mathcal{V}_{eff}^{(T)} \propto \left\{\binom{d}{K}\left[1 + \frac{\alpha K}{(1 - \alpha)(d - K)}\right]^K\right\}^T,$$

(6)

where each time step contributes multiplicatively to the total space due to the evolving trace states. Such expansion directly translates to enhanced generalization capability, as the SNNs can develop more specialized and nuanced temporal feature detectors.

In spiking networks, such dynamic trace-based modulation not only increases feature dimensionality but also enables the model to encode fine-grained temporal correlations across inputs. Therefore, the stochastic selection via RTK-WTA enhances generalization by both increasing representational diversity and enriching the spatiotemporal discriminative capacity.

Generalization capability can be theoretically linked to the minimal inter-class margin in the feature representation space. Consider $n$ different classes, represented by class-specific trace vectors $W_1, ..., W_n$ after training. The minimal inter-class margin $d_W$ between any pair of classes $(W_i, W_j)$ is defined as

$$d_W = \min_{i \neq j} |W_i - W_j|_2. \tag{7}$$

With the temporal RTK-WTA mechanism, the effective inter-class margin $d_W^{RTK}$ is scaled by the increased feature space volume. Specifically:

$$d_W^{RTK} \propto \left( \frac{\mathcal{V}_{eff}^{(T)}}{n} \right)^{\frac{1}{KT-1}}, \tag{8}$$

which is significantly larger compared to deterministic Top-K scenarios, thus substantially reducing the generalization error.

Thus, by integrating trace-based temporal dynamics and randomized neuron selection, the proposed RTK-WTA SNNs not only increase the effective feature space but also improve the robustness of learning dynamics, jointly contributing to enhanced generalization capabilities of SNNs.

## 3 RESULTS

In this section, we evaluate the performance of the proposed RTK-WTA SNNs for continual learning tasks. The ablation studies, robustness validation, and neuronal selective activation analysis are conducted to validate the effectiveness of our model.

### 3.1 PERFORMANCE COMPARISON

Our evaluation framework incorporates multiple baselines to enable meaningful architectural comparisons, including Flymodel Shen et al. (2021), SDMLP Bricken et al. (2023), EWC Bricken et al. (2023); Kirkpatrick et al. (2017) and SA-SNN Shen et al. (2024). The SDMLP baseline provides a particularly insightful reference point - as an ANN with fully connected Top-K sparse activation, it shares our RTK-WTA SNN's core architectural principles while operating in the non-spiking domain. This structural similarity allows us to isolate the benefits of our spiking temporal dynamics while avoiding the confounding effects of convolutional or attention operations. The validity of SDMLP as a comparative benchmark is further supported by its established use in continual learning literature Van de Ven et al. (2020).

We comprehensively evaluate the proposed RTK-WTA mechanism against state-of-the-art continual learning methods under similar archi-

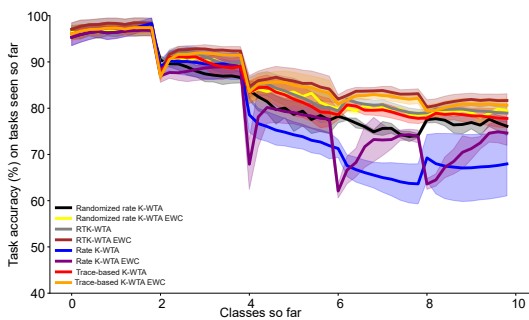

Figure 2: The performance comparison between our model and some other baseline models on the splitCIFAR10 dataset. The horizontal axis represents the number of classes encountered as tasks increment, and model performance is measured using accuracy.

tectures across three benchmark datasets: splitMNIST, splitCIFAR10, and splitCIFAR100. As shown in Table 1, our method achieves superior performance in both standalone and EWC-enhanced configurations, demonstrating its effectiveness in mitigating catastrophic forgetting while maintaining computational efficiency.

**Standalone Performance.** The proposed RTK-WTA mechanism outperforms all baseline methods across all datasets. On splitMNIST, it achieves 60.37% accuracy, surpassing the best baseline (trace-based SA-SNN: 60.06%) by 0.31%. The advantage becomes more pronounced on complex tasks: for splitCIFAR100, RTK-WTA achieves 32.91% accuracy, exceeding the second-best method (Trace-based SA-SNN: 22.86%) by 10.05%. This aligns with our theoretical analysis in the method section:

Table 1: Performance comparison among different models with the similar architectures.

| Methods | splitMNIST | splitCIFAR10 | splitCIFAR100 |
|---|---|---|---|
| None | 19.96 (+/-0.01) | 25.29 (+/-1.97) | 4.28(+/-0.38) |
| Joint | 97.72 (+/-0.04) | 92.19 (+/-0.08) | 65.28(+/-0.15) |
| EWC-SNN | 19.89 (+/-0.00) | 30.04 (+/-2.65) | 3.93(+/-0.24) |
| MAS-SNN | 19.91 (+/-0.01) | 30.44 (+/-2.60) | 3.09(+/-0.45) |
| SDMLP | 46.67 (+/-3.92) | 73.27 (+/-1.28) | - |
| FlyModel | 76.97 (+/-1.26) | 70.09 (+/-0.51) | 17.25 (+/-0.42) |
| Rate-based SA-SNN | 50.22 (+/-0.91) | 76.88 (+/-2.12) | 21.37(+/-0.77) |
| Trace-based SA-SNN | 60.06 (+/- 2.16) | 77.73 (+/-1.95) | 22.86 (+/-0.65) |
| Randomized Rate K-WTA | 48.15 (+/-2.02) | 76.11 (+/-0.78) | 20.76 (+/-1.51) |
| **RTK-WTA** | **60.37 (+/- 2.78)** | **78.37 (+/- 0.99)** | **32.91 (+/-0.46)** |
| SDMLP + EWC | 79.61 (+/- 2.46) | 78.64 (+/- 0.30) | 21.31 (+/-0.72) |
| SA-SNN + EWC | 82.18 (+/- 1.14) | 80.39 (+/- 1.84) | 36.47 (+/- 2.13) |
| Randomized Rate K-WTA + EWC | 80.12 (+/- 0.95) | 79.97 (+/-1.01) | 22.21(+/-2.37) |
| **RTK-WTA + EWC** | **85.25 (+/- 0.67)** | **80.56 (+/- 0.66)** | **41.46 (+/- 0.23)** |

the expanded feature space volume in equation 6 enables better preservation of task-specific features through stochastic neuron selection. Notably, rate-based approaches (*e.g., Random Rate-based Top K*) underperform by 12.2% on splitCIFAR100, confirming the critical role of temporal dynamics in trace-based methods for high-dimensional tasks.

**Integration with EWC.** When combined with EWC, our method (Random Trace-based TopK + EWC) achieves state-of-the-art results: 85.25% on splitMNIST (+3.07% over SA-SNN + EWC) and 41.46% on splitCIFAR100 (+5.0% over SA-SNN + EWC). This synergy arises from two complementary mechanisms: (1) EWC protects critical synaptic weights identified by Fisher information, while (2) RTK-WTA SNN's stochastic masking (equation 3) prevents overfitting to noisy or task-specific connections. The gradient noise variance $\text{Var}(\Delta\theta_{\text{noise}}) \propto \frac{\alpha(1-\alpha)}{K(d-K)}$ further regularizes optimization, enabling robust learning across tasks. In contrast, deterministic Top-K variants (e.g., Trace-based SA-SNN + EWC) exhibit degraded performance on splitCIFAR100 (36.47% vs. 41.46%), as their fixed neuron selection amplifies weight interference.

Moreover, the performance gap between RTK-WTA and baselines correlates with our margin analysis: the inter-class margin $d_W^{\text{RTK}}$ expands by $\sim\sqrt{\frac{\alpha K}{(1-\alpha)(d-K)}}$ compared to deterministic Top-K. This explains why RTK-WTA achieves 32.91% on splitCIFAR100 versus 22.86% for Trace-based SA-SNN – the larger margin better separates 100-class representations.

In addition, when benchmarked on splitCIFAR10, the class accuracy curve of our RTK-WTA in Figure 2 shows the robust retention of the learned knowledge in the incremental learning process, and shows the competitive performance of our model compared with the baseline. As time goes by, it always maintains the recognition ability of all learning classes.The sustained high accuracy across all learned classes (ranging from initial to most recent tasks) suggests effective prevention of catastrophic forgetting, while the competitive new task learning speed indicates maintained plasticity.

## 3.2 INFLUENCE OF DIFFERENT LEVELS OF RANDOMNESS IN RTK-WTA

The impact of the randomness coefficient $\alpha$ in our RTK-WTA SNNs is systematically evaluated across four configurations: standalone/EWC-enhanced randomized rate-based and trace-based models. As shown in Figure 3, all methods exhibit a consistent trend: performance first improves with increasing $\alpha$, peaks at $\alpha = 0.1$, then declines as $\alpha$ exceeds this optimal value. This non-monotonic behavior validates our theoretical framework connecting controlled randomness to spatiotemporal feature learning in SNNs.

**Performance Under Different Levels of Randomness.** The performance peak at $\alpha = 0.1$ aligns with our derived feature space expansion. When $\alpha = 0.1$, the expansion factor $\alpha K/(1-\alpha)(d-K)$ reaches an optimal balance: sufficient diversity to capture spatiotemporal patterns (improving accuracy by 1.3% over $\alpha = 0$ for Random-k_step), yet preserving the core Top-K selection mechanism.

Table 2: The robustness comparison of different models on different datasets.

| CIFAR10 dataset | | | | | | |
|---|---|---|---|---|---|---|
| Noise levels | 0 | 0.1 | 0.2 | 0.5 | 0.6 | 0.8 |
| Rate-based SA-SNN | 76.88 | 69.59 | 66.80 | 65.40 | 43.58 | 37.77 |
| Rate-based SA-SNN + EWC | 79.86 | 75.10 | 72.14 | 69.75 | 57.80 | 53.65 |
| Trace-based SA-SNN | 77.73 | 76.76 | 76.73 | 74.66 | 61.85 | 58.95 |
| Trace-based SA-SNN + EWC | 80.39 | 79.88 | 79.45 | 78.86 | 70.84 | 59.69 |
| Randomized Rate K-WTA | 76.11 | 74.64 | 73.3 | 66.42 | 65.45 | 54.28 |
| Randomized Rate K-WTA + EWC | 79.97 | 77.91 | 74.97 | 71.26 | 70.87 | 62.77 |
| RTK-WTA | 78.37 | 76.92 | 77.25 | 75.78 | 66.6 | 61.36 |
| **RTK-WTA + EWC** | **80.56** | **80.12** | **80.14** | **77.96** | **73.06** | **68.85** |

| CIFAR100 dataset | | | | | | |
|---|---|---|---|---|---|---|
| Noise levels | 0 | 0.1 | 0.2 | 0.5 | 0.6 | 0.8 |
| Rate-based SA-SNN | 21.37 | 20.03 | 19.11 | 14.36 | 11.61 | 9.16 |
| Rate-based SA-SNN + EWC | 21.88 | 20.36 | 19.65 | 16.75 | 13.59 | 9.88 |
| Trace-based SA-SNN | 22.86 | 23.10 | 21.06 | 16.12 | 12.28 | 9.23 |
| Trace-based SA-SNN + EWC | 36.47 | 33.56 | 31.24 | 26.32 | 17.63 | 10.56 |
| Randomized Rate K-WTA | 20.76 | 20.15 | 19.45 | 16.21 | 13.93 | 9.82 |
| Randomized Rate K-WTA + EWC | 22.21 | 20.45 | 20.56 | 17.31 | 16.04 | 10.06 |
| RTK-WTA | 32.91 | 31.49 | 31.14 | 28.63 | 23.19 | 13.34 |
| **RTK-WTA + EWC** | **41.46** | **39.67** | **37.64** | **32.42** | **26.37** | **17.69** |

When $\alpha > 0.1$, performance declines sharply (e.g., -14.47% drop for Random-k_step at $\alpha = 0.5$). This occurs because excessive randomness violates the theoretical condition for effective K-WTA operation. At $\alpha = 0.5$, nearly half of the selected neurons come from non-Top-K regions, disrupting the temporal coherence resulting the irregular effective neuron selection.

**Performance Changes Under Different Randomness with EWC.** The EWC-enhanced variants exhibit flatter degradation curves, with our RTK-WTA SNNs with EWC maintaining 76.31% accuracy even at $\alpha = 0.5$ (-4.25% from peak). This demonstrates EWC's complementary role: by constraining Fisher-important weights, it mitigates the destabilizing effect of high $\alpha$, though cannot fully compensate for fundamental K-WTA mechanism breakdown. The

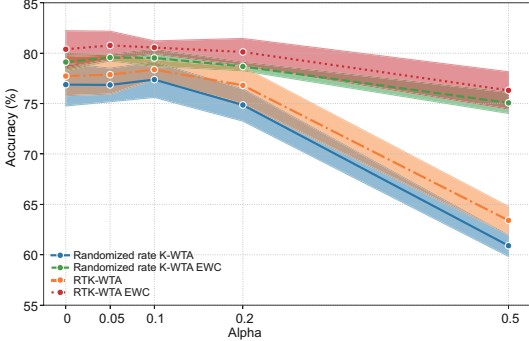

Figure 3: The influence of different levels of randomness in RTK-WTA SNNs.Comparative analysis reveals that all methods attain superior performance when $\alpha$ is set to 0.1.

gradient noise variance of $\text{Var}(\Delta\theta_{\text{noise}}) \propto \frac{\alpha(1-\alpha)}{K(d-K)} \sum_{t,i} \left( \frac{\partial L}{\partial tr_i[t]} \frac{\partial tr_i[t]}{\partial \theta} \right)^2$ remains beneficial up to $\alpha = 0.2$, beyond which the $\alpha(1-\alpha)$ term decreases while misselection dominates.

Therefore, our results establish $\alpha = 0.1$ as the empirically optimal setting, achieving 1.64% improvement over deterministic Top-K ($\alpha = 0$) and about $5.8\times$ slower degradation than rate-based methods at $\alpha = 0.5$. That balances the dual needs of feature diversity and temporal stability, making our WTA-RTK SNNs adaptable to various neuromorphic scenarios without parameter tuning overhead.

## 3.3 ROBUSTNESS OF RTK-WTA SNNS

We evaluate the robustness of the proposed SNN with RTK-WTA SNNs against deterministic Top-K variants under two scenarios: (1) *training with noise and testing without noise*, and (2) *training without noise and testing with noise*. Noise is injected during training by randomly corrupting a fraction (*Noise Level*) of the Top-K neuronal connections, simulating unstable synaptic transmission.

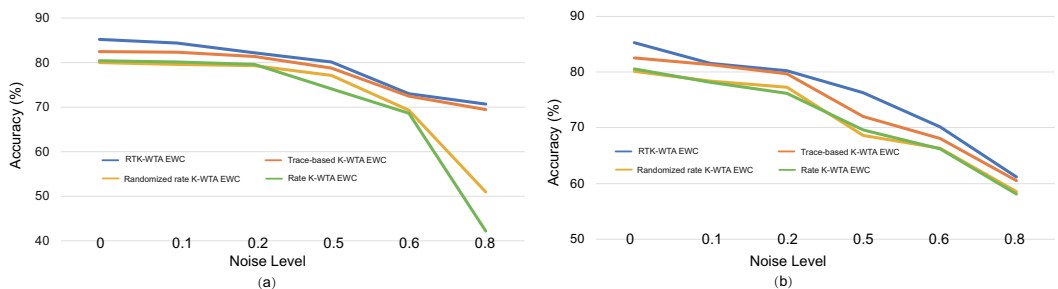

Figure 4: The robustness of the proposed RTK-WTA SNNs. (a) Adding different levels of noises on training dataset. (b) Adding different levels of noises on test dataset.The results show that our RTK-WTA+EWC consistently outperforms all other methods across all tested noise levels.

The results on splitMNIST under incremental class learning demonstrate the superiority of RTK-WTA in handling both training and testing noise.

As shown in the Figure 4 (a), RTK-WTA SNNs with EWC exhibit significantly smaller performance degradation as noise increases. For instance, our RTK-WTA SNNs with EWC decline by only 14.56% (85.25% → 70.69%) from Noise=0 to 0.8, whereas deterministic trace-based K-WTA with EWC without randomness drops by 12.91% (82.47% → 69.56%). This resilience stems from the expanded feature space volume, which enables RTK-WTA to learn noise-invariant representations by stochastically masking neurons during training. The trace decay dynamics in RTK-WTA SNNs further stabilize temporal integration, reducing sensitivity to corrupted connections. In contrast, rate-based K-WTA SNNs with EWC SNNs suffer catastrophic failure at high noise (42.19% at Noise=0.8), as static rate coding lacks temporal adaptability.

Figure 4 (b) reveals that RTK-WTA SNNs generalize better to unseen noise. our RTK-WTA SNNs with EWC maintain a 24.04% accuracy margin over trace-based K-WTA SNNs with EWC at Noise=0.8 (61.21% vs. 60.48%), despite both using trace-based selection. This advantage arises from gradient noise induced by RTK-WTA, where the variance term promotes convergence to flatter minima, inherently robust to input perturbations. The stochastic masking during training mimics test-time noise, effectively acting as implicit regularization. Meanwhile, deterministic trace-based K-WTA SNNs with EWC without randomness overfit to clean training data, leading to sharper minima and faster performance collapse under noise.

Furthermore, we conduct robustness validation experiments under increasing noise perturbation levels on CIFAR-10, CIFAR100 and Tiny ImageNet (See supplementary materials). Only the results of adding noise to the training set are displayed in Table 2. These results validate that RTK-WTA SNNs consistently outperform deterministic WTA baselines and show higher resilience across all tested noise levels.

The results align with our theoretical analysis that RTK-WTA SNNs' stochastic neuron selection expands the effective feature space and enhances the robustness of SNNs for continual learning. Combined with temporal trace dynamics, this mechanism mitigates the double curse of noise—corrupted training connections and shifted test distributions. Practically, RTK-WTA SNNs requires no additional modules or hyperparameters beyond $\alpha$, making it scalable for neuromorphic applications.

### 3.4 NEURONAL SELECTIVELY ACTIVATION ANALYSIS.

**Visualization of the Neuronal Distribution.** Analysis of neuronal selectivity distributions in Figure 5 reveals that while the rate-based K-WTA SNNs effectively activate task-relevant neurons across all categories, they require a substantially larger population of highly selective neurons. In contrast, the trace-based K-WTA SNNs achieve superior performance with fewer specialized neurons, demonstrating significantly improved neural resource utilization efficiency by considering the temporal dynamics. Furthermore, our RTK-WTA SNNs maintain remarkable consistency with trace-based K-WTA SNNs in the neuronal activation patterns, indicating that the incorporation of stochastic mechanisms not only preserves balanced neuronal selectivity distributions but also enhances performance while maintaining comparable resource efficiency.

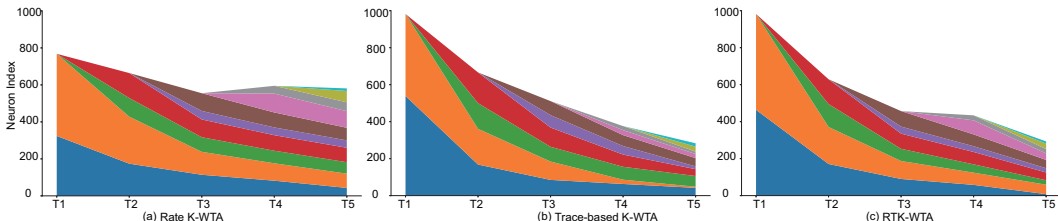

Figure 5: Neuronal selectivity analysis among different SNNs, each color represents the activated neurons associated with one of the 10 class labels.

**Quantitative Analysis of Neuronal Activation Coverage.** A quantitative metric termed Neuronal Activation Coverage is proposed, which represents the proportion of neurons that exhibit significant activation (i.e., an activation count exceeding a predefined threshold) throughout the entire continual learning process. We computed the neuronal activation coverage on the CIFAR-10 classification task, using an average firing rate of at least 0.3 spikes per sample as the threshold to determine significant activation for each neuron. The results show that RTK-WTA significantly increases neuronal utilization across the network (45.2% vs. 28.5% for deterministic Trace-based K-WTA). This indicates that a more diverse set of neurons participates in feature representation for different tasks, rather than relying on a fixed subset of repeatedly activated neurons.

## 4 CONCLUSION

In this work, we introduced Randomized Temporal Winner-Take-All (RTK-WTA), a biologically inspired selective activation mechanism for spiking neural networks in continual learning. By leveraging temporally accumulated neuronal traces and probabilistic competition, RTK-WTA enables context-aware, sparse activation that dynamically adapts to evolving task sequences. This mechanism not only expands the effective spatiotemporal feature space but also enhances representational diversity and robustness, allowing the network to better mitigate catastrophic forgetting. Empirical results across multiple benchmarks, including SplitMNIST, SplitCIFAR-10, and SplitCIFAR-100, show that our approach outperforms traditional deterministic Top-K baselines by a significant margin while demonstrating strong resilience to noise and task interference. The success of RTK-WTA highlights the potential of integrating temporal selectivity with stochastic activation to achieve biologically plausible and scalable lifelong learning in neuromorphic systems. Restricted to relatively shallow networks in its current form, extending this framework to deeper architectures to enable robust representation learning for complex continual tasks is a priority for future work.

## ACKNOWLEDGEMENTS

This work was supported by National Key Research and Development Program of China (Grant No.2023YFB4705502), National Natural Science Foundation of China under Grant (No.62306274, 62476035, 62436005, 62473303, U24B20140), China Postdoctoral Science Foundation under Grant Number (No. GZB20250394, 2025T180427, 2025M771543).

## ETHICS STATEMENT

All authors disclosed no relevant relationships. No potential conflict of interest was reported by the authors. This article does not contain any studies with human participants or animals performed by any of the authors, therefore ethical approval was not required.

## REPRODUCIBILITY STATEMENT

Efforts to ensure reproducibility include: Detailed experimental configurations, model architectures, and hyperparameters are provided in the appendix. The source code will be released publicly upon acceptance of this paper.

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

# A  APPENDIX

## A.1  RELATED WORK

More detailed related works are summarized as follows:

**SNNs for Continual Learning** The pursuit of continual learning in SNNs draws inspiration from biological systems that demonstrate lifelong learning capabilities Kandel & Hawkins (1992). Recent advances include Antonov et al. (2022)'s unsupervised framework combining Langevin dynamics with STDP for synaptic importance estimation and Skatchkovsky et al. (2022)'s Bayesian online learning rule for dynamic environments. Evolutionary strategies were employed by Hammouamri et al. (2022) to optimize auxiliary networks for dynamic threshold adjustment, while Tadros et al. (2022) introduced local plasticity with coding conversion to mitigate biases. These approaches benefit from SNNs' inherent temporal dynamics Maass (1997) and energy efficiency Pei et al. (2019), though challenges remain in achieving biological-like flexibility Kudithipudi et al. (2022).

Winner-take-all circuits in SNNs implement biological inhibition principles observed in neural systems Lin et al. (2014); Douglas & Martin (2004). Lynch et al. (2019) developed a stochastic implementation achieving O(1) convergence through adaptive inhibition, while Su et al. (2019) derived theoretical limits for k-WTA circuits. The biological plausibility is further supported by Shukla et al. (2019)'s work combining hard WTA for Gibbs sampling and soft WTA for approximate inference. Challenges in temporal implementation have prompted solutions using STDP rules Markram et al. (1997) and trace variables Morrison et al. (2008), addressing issues like homogeneous spike patterns Shen et al. (2021) and neuron death Ahmad & Scheinkman (2019). Notably, the sparse activation property of SNNs based on temporal trace-based K-WTA Shen et al. (2024) naturally reduces memory interference and enhances the feature representation ability of SNNs, resembling biological context gating mechanisms Kay & Laurent (1999). These developments mirror biological neuronal variability Izhikevich (2004) while advancing neuromorphic computing.

In addition to architectural or activation-level mechanisms, recent works have examined optimization and initialization strategies to stabilize deep SNN training Ding et al. (2025).

**Selective Activation Mechanisms** Biological selective activation operates through multi-scale mechanisms. At the circuit level, Douglas & Martin (2004)'s soft WTA model demonstrates how pyramidal-interneuron competition enables input-adaptive selection, while Buzsáki (2010)'s reader-actor framework explains assembly selection via dynamic synapsembles. Molecular-level evidence comes from Imokawa & Brockes (2003)'s work on thromboxane activation in regeneration. These processes are modulated by context-dependent mechanisms Levinson et al. (2020) that inspire artificial implementations Kudithipudi et al. (2022). Crucially, biological systems achieve this through sparse activation patterns Stevens (2015) and specialized inhibitory circuits Lin et al. (2014), principles now informing SNN designs Shen et al. (2023).

Biological selective activation principles have inspired three technical implementations in neural networks. Wang et al. (2023)'s AMU with SAPW/SAPI strategies reduces computation while maintaining accuracy, paralleling biological energy conservation Imam & Cleland (2020). Geifman & El-Yaniv (2017)'s rejection mechanism enables risk-controlled prediction, akin to biological decision circuits Buzsáki (2010). Hardware solutions like Wang et al. (2020)'s self-selective memory devices achieve efficiency gains while avoiding common pitfalls like neuron death Fedus et al. (2022). These artificial systems complement conventional approaches Kirkpatrick et al. (2017); Rebuffi et al. (2017) by incorporating biological constraints McCloskey & Cohen (1989), though challenges remain in fully capturing temporal dynamics Dan & Poo (2004).

## A.2  SUPPLEMENTARY EVALUATION OF NOISE ROBUSTNESS

Table 3: The noise experimental results on Tiny-imageNet dataset.

| Tiny ImageNet dataset | | | | |
|---|---|---|---|---|
| Noise levels | 0 | 0.1 | 0.2 | 0.5 |
| Rate-based SA-SNN | 24.71 | 24.15 | 23.54 | 20.34 |
| Rate-based SA-SNN + EWC | 28.12 | 27.68 | 26.78 | 21.50 |
| Trace-based SA-SNN | 31.25 | 30.77 | 30.59 | 22.03 |
| Trace-based SA-SNN + EWC | 35.50 | 33.89 | 32.97 | 27.00 |
| Randomized Rate K-WTA | 29.06 | 28.36 | 27.09 | 20.80 |
| Randomized Rate K-WTA + EWC | 31.49 | 30.24 | 26.8 | 22.23 |
| RTK-WTA | 37.42 | 36.87 | 37.34 | 35.07 |
| **RTK-WTA + EWC** | **43.17** | **40.19** | **39.22** | **36.54** |

The supplementary experimental results of the noise robustness on Tiny-ImageNet dataset are supplemented as shown in Table 3.

## A.3   ENERGY ESTIMATES

In SNNs, fewer spikes generally equate to lower energy usage. We provide a quantitative comparison of energy consumption across different models when performing inference on a single CIFAR-100 image with a time window of 16 steps.

Table 4: Energy consumption comparison between SNN and ANN models

| SNN Metric | RTK-WTA | Randomized Rate K-WTA | Trace-based SA-SNN |
|---|---|---|---|
| Total Spike Activity | 2.27E-08 J | 2.18E-08 J | 2.27E-08 J |
| TOP-K Selection | 6.72E-09 J | 1.70E-10 J | 6.72E-09 J |
| Randomization Control | 6.40E-09 J | 2.00E-09 J | 0 |
| Total Energy (per image) | 3.58E-08 J | 2.40E-08 J | 2.94E-08 J |

| Rate-based SA-SNN | Pure SNN | - | SDMLP (ANN) | ANN Metric |
|---|---|---|---|---|
| 2.18E-08 J | 2.07E-08 J | - | 6,344,000 ops | MACs |
| 1.70E-10 J | 0 | - | 12,722,590 ops | FLOPs |
| 0 | 0 | - | 2.92E-05 J | Total |
| 2.20E-08 J | 2.07E-08 J | - | 2.92E-05 J | Total Energy |

As Table 4 shows, RTK-WTA consumes only 3.58E-08 J, which is comparable to the SNN baselines, such as Trace-based SA-SNN (2.94E-08 J). More importantly, the energy cost is orders of magnitude lower than standard ANN models (e.g., SDMLP: 2.92E-05 J), indicating the approach remains highly energy-efficient despite incorporating trace dynamics.

## A.4   ABLATION EXPERIMENT

We conduct systematic ablation studies to precisely disentangle the contribution of each component. The results are summarized in the Table 5 (all methods use EWC).

(a) Role of Stochastic Selection: Comparing RTK-WTA with its deterministic version (setting $\alpha$=0), the removal of stochasticity leads to a performance drop, particularly evident on the complex

Table 5: Ablation experiment results on different datasets.

| Method | MNIST | CIFAR10 | CIFAR100 |
|---|---|---|---|
| RTK-WTA (Full Model) | 85.25 | 80.37 | 41.46 |
| (a) - probabilistic selection | 82.18 | 80.09 | 36.47 |
| (b) - trace | 80.12 | 80.02 | 22.21 |
| - probabilistic selection & trace | 80.22 | 79.86 | 19.88 |

Table 6: The supplementary experimental results on incremental MNIST dataset.

| Incremental MNIST dataset | |
| --- | --- |
| Method | Avg_acc |
| Rate-based SA-SNN | 40.9 |
| Trace-based SA-SNN | 40.3 |
| Rate-based SA-SNN + EWC | 69.4 |
| Trace-based SA-SNN + EWC | 78.9 |
| Randomized Rate K-WTA | 44.2 |
| RTK-WTA | 45.2 |
| Randomized Rate K-WTA + EWC | 72.5 |
| **RTK-WTA + EWC** | **79.5** |

SplitCIFAR100 with a decrease of 5.0%. This confirms that stochastic selection is crucial for mitigating interference between complex tasks.

(b) Role of Trace Modulation: Replacing the Trace-based selection with Rate-based selection causes a sharp performance decline, with a dramatic drop of 19.25% on SplitCIFAR100. This firmly establishes the critical role of trace modulation in capturing temporal dynamics and enhancing continual learning performance.

The ablation studies clearly quantify the independent contribution of each component and their synergistic effect, providing strong support for our methodological design.

## A.5 EVALUATION IN CLASS-INCREMENTAL SCENARIOS

We conduct additional experiments on class-incremental continual learning, which presents a more realistic and demanding setting than the task-incremental scenario originally reported. Our results on incremental MNIST (iMNIST), as shown in Table 6, demonstrate that RTK-WTA maintains its superiority over deterministic K-WTA baselines even under Class-IL conditions, achieving an accuracy improvement of approximately 4.9%. Furthermore, when combined with EWC, our method continues to deliver the best performance. These findings strongly support the generalizability and robustness of the RTK-WTA mechanism across different continual learning paradigms.

## A.6 T-SNE VISUALIZATION ANALYSIS.

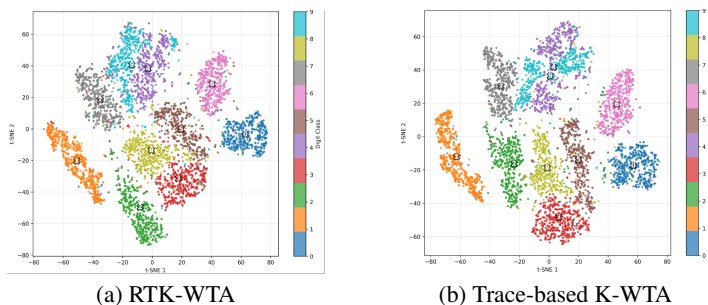

(a) RTK-WTA        (b) Trace-based K-WTA

Figure 6: t-SNE visualization results comparing RTK-WTA and trace-based K-WTA

We perform t-SNE visualizations of the feature space for CIFAR-10's 10-class classification task during testing. Figure 6 (a) shows class clusters for RTK-WTA, while Figure 6 (b) displays those for the deterministic K-WTA baseline.The visualization clearly demonstrates that RTK-WTA forms more discriminative and well-separated class clusters in the feature space, exhibiting superior feature separation compared to the deterministic K-WTA baseline, which shows partial distribution aliasing.

Table 7: Experimental results on Tiny ImageNet dataset.

| Tiny ImageNet dataset | |
|---|---|
| Method | Acc (%) |
| Rate-based SA-SNN | 19.26 |
| Trace-based SA-SNN | 23.22 |
| Rate-based SA-SNN + EWC | 17.83 |
| Trace-based SA-SNN + EWC | 28.78 |
| Randomized Rate K-WTA | 22.25 |
| RTK-WTA | 26.42 |
| Randomized Rate K-WTA + EWC | 25.56 |
| **RTK-WTA + EWC** | **32.94** |

## A.7 SUPPLEMENTARY END-TO-END EXPERIMENTS

To comprehensively evaluate the scalability and generalization capability of our method, we conducted additional experiments using an end-to-end SNN-ViT architecture on the more challenging TinyImageNet dataset. With 200 classes and higher image resolution, TinyImageNet provides a significantly larger and more complex benchmark for continual learning compared to the CIFAR series.

The results presented in Table 7 confirm the RTK-WTA mechanism can be effectively integrated into deep spiking neural networks, achieving a performance improvement of approximately 4.2% over the deterministic Top-K baseline. Moreover, RTK-WTA consistently outperforms all baseline methods in this challenging setting. Notably, when combined with EWC, RTK-WTA attains an accuracy of 32.94%, substantially surpassing the strongest trace-based deterministic baseline (Trace-based SA-SNN + EWC at 28.78%).

These findings strongly validate the excellent scalability and generalization capacity of the RTK-WTA mechanism, confirming its efficacy in handling large-scale, multi-class continual learning tasks.

## A.8 SUPPLEMENTARY THEORETICAL ANALYSIS

We analyze the information gain introduced by the stochasticity in the RTK-WTA selection, while holding the trace mechanism constant.

To provide a more formal foundation for the benefit of random trace-based top-k selection in RTK-WTA, we present a derivation based on mutual information (MI) under entropy constraints.

In detail, we further analyze the information gain introduced specifically by stochasticity in the RTK-WTA selection, holding the trace mechanism constant.

Let $X \in \mathbb{R}^d$ be input feature vector of traces in our RTK-WTA. $Y$ denotes the output label or task. $S \subset \{1, \ldots, d\}$ denotes selected neuron indices (top-k), while $Z = X_S$ are the activations of the selected neurons.

We analyze the mutual information $I(Z; Y)$, which quantifies how much information the selected neurons convey about the task output. We apply a standard lower bound $I(Z; Y) \geq \mathbb{E}_{S|X} \log q(Y|Z) + H(Y)$ where $q(Y|Z)$ is a variational approximation of the true posterior.

This implies maximizing the likelihood of correct prediction encourages high mutual information between selected features and output.

Instead of deterministic top-k selection, since we design the RTK-WTA SNN with a stochastic selector $P(S|X)$, the mutual information under stochastic selection becomes $I(X_S; Y) = \mathbb{E}_{S \sim P(S|X)}[I(X_S; Y)]$.

We optimize the selector via: $\max_{P(S|X)} \mathbb{E}_S[I(X_S; Y)] - \lambda H(P(S|X))$.

This balances information preservation with selection diversity (controlled via entropy). RTK-WTA approximates this entropy-constrained sampler. Let:

$$P(i \in S|X) = \begin{cases} \text{SoftTopK}_\alpha(i) & \text{if } i \in \text{Top-d} \\ 0 & \text{otherwise} \end{cases}$$

where $\alpha$ controls the sampling temperature. When $\alpha \to 0$, it satisfies uniform random selection (high entropy) while $\alpha \to \infty$, it satisfies the deterministic top-k (zero entropy). Thus, RTK-WTA enforces selectivity while promoting diversity across tasks.

Let $Z_{\text{RTK}} \sim P(S|X)$ represent RTK-WTA selection, and $Z_K = \text{top-K}(X)$ represent the deterministic K-WTA.

Then $I(Z_{\text{RTK}}; Y) > I(Z_K; Y)$, under continual learning conditions, where overlapping tasks create interference.

RTK-WTA enables richer representational modes and higher task specificity through stochasticity.

Therefore, the proposed RTK-WTA SNNs can be interpreted as an entropy-regularized top-k mechanism that increases mutual information by expanding the selection space over time, avoiding representational collapse and improving robustness to interference.

## A.9 USE OF LLMs

Large Language Models(LLMs) were used solely to assist with polishing the text.

## A.10 CODE OF ETHICS AND ETHICS STATEMENT

The research conducted in the paper conform, in every respect, with the ICLR Code of Ethics https://iclr.cc/public/CodeOfEthics.

## A.11 EXPERIMENTAL SETUP

All experiments were conducted on a high-performance computing cluster with uniform hardware/software configurations to ensure strict reproducibility. The specific configuration is shown in the Table 8.

Table 8: Hardware and Software Environment

| Item | Configuration |
|---|---|
| Operating System | Linux (Ubuntu 22.04 LTS) |
| CPU | Intel Xeon Gold 6248R (3.0GHz, 48 cores) |
| GPU | NVIDIA RTX 4090 (24GB VRAM) |
| Python | 3.10.16 |
| PyTorch | 2.6.0+cu118 |
| CUDA | 11.8 |
| cuDNN | 9.1.0 |

## A.12 MODEL TRAINING CONFIGURATION

Our training strategy combines unsupervised RTK-WTA pre-activation with gradient-based updates (when supervised labels are available), and integrates EWC for synaptic consolidation. We use surrogate gradient descent for training spike-based modules, following standard SNN training practices in Wu et al. (2018). The detailed architectural diagrams, training schedules, and evaluation protocols are described as follows:

For MNIST experiments, we employed a two-layer fully connected network architecture, consisting of an input layer followed by a hidden layer with 1000 ReLU-activated nodes, with task-specific output layers (one per binary classification task, totaling 5 tasks) trained end-to-end.

For CIFAR10 and CIFAR100 experiments, we first extracted image embeddings using ResNet-18 backbone, then fed them into a two-layer fully connected network. The CIFAR10 model featured

binary classification output layers (5 tasks total), while the CIFAR100 model utilized a larger hidden layer (2000 nodes) and 4-class output layers (25 tasks total). We conducted two experiments on the Tiny ImageNet dataset: the first employed a pre-trained ViT model for feature extraction, where the resulting embeddings were processed by a two-layer fully-connected neural network to enable comparative analysis of different methods; the second adopted an end-to-end approach by constructing a custom ViT-style feature extractor that utilizes convolutional layers for 16×16 patch embedding, incorporates learnable [CLS] tokens and positional encodings, processes features through a 4-layer Transformer encoder (with 3 attention heads per layer, 384 feed-forward dimension, and 0.1 dropout rate), and ultimately outputs normalized [CLS] token features (192-dimensional) - with subsequent classifier training following the same procedure as before to enable simultaneous optimization of both feature extractor and classifier. Various methods were implemented in combination with either EWC regularization or sparsity-inducing mechanisms (such as Top-K Trace) to alleviate catastrophic forgetting during continual learning.

Besides, for the MNIST experiments, we directly used raw images as input to train and compare different continual learning methods under a sequential task paradigm. Similarly, the end-to-end TinyImageNet experiments processed raw images through an instantiated End2EndContinualLearner (comprising both feature extractor and classifier), following the same sequential task training procedure. In contrast, the CIFAR10/100 and TinyImageNet embeddings experiments adopted a two-phase training pipeline: (1) a feature extraction phase where pretrained models generated image embeddings, followed by (2) a classifier training phase with frozen feature extractors to enable continual learning in the fixed representation space.

The parameter settings of our proposed method are detailed in Figure 9 and Figure 10. Among them,the $\lambda$ parameter controls the weight of the EWC loss term in the total loss function, where lower $\lambda$ values permit greater parameter updates. The $\beta$ parameter regulates the smoothing intensity of the diagonal elements in the Fisher information matrix.The Adaptation Speed ($\tau$) controls the decay rate of neuron threshold adjustments, with higher values resulting in slower adaptation, which is tuned to balance plasticity and stability.The L2-Norm scale($\gamma$) is used to control the scaling magnitude after weight normalization, balancing the weight magnitudes across neural network layers.

Table 9: Training Parameters for RTK-WTA+EWC

| Parameter | MNIST | CIFAR10 | CIFAR100 |
|---|---|---|---|
| Iterations | 20,000 | 20,000 | 8,000 |
| Batch Size | 64 | 512 | 256 |
| Learning Rate ($\eta$) | 0.05 | 0.05 | 0.08 |
| $\lambda$(EWC) | 400 | 200 | 100 |
| $\beta$(EWC) | 0.08 | 0.08 | 0.08 |
| Adaptation Speed ($\tau$) | $1.0 \times 10^6$ | $2.0 \times 10^6$ | $2.0 \times 10^6$ |
| L2-Norm Scale ($\gamma$) | 1.00 | 0.25 | 0.25 |
| Optimizer | SGD | SGD | SGD |

Table 10: Training Parameters for Randomized Rate K-WTA+EWC

| Parameter | MNIST | CIFAR10 | CIFAR100 |
|---|---|---|---|
| Iterations | 20,000 | 20,000 | 8,000 |
| Batch Size | 64 | 512 | 256 |
| Learning Rate ($\eta$) | 0.05 | 0.05 | 0.08 |
| $\lambda$(EWC) | 400 | 200 | 100 |
| $\beta$(EWC) | 0.08 | 0.08 | 0.08 |
| Adaptation Speed ($\tau$) | $1.5 \times 10^6$ | $2.0 \times 10^6$ | $2.0 \times 10^6$ |
| Optimizer | SGD | SGD | SGD |

