# OpenReview forum: "Robust Selective Activation with Randomized Temporal K-Winner-Take-All in Spiking Neural Networks for Continual Learning"
_ICLR.cc/2026/Conference — ICLR 2026 Poster_

### Official Review · Reviewer_27Cb · 2025-10-28

**Soundness:** 3
**Presentation:** 2
**Contribution:** 3
**Rating:** 6
**Confidence:** 4

**Summary:**

This paper introduces Randomized Temporal K-winner-take-all (RTK-WTA), a continual learning (CL) framework for Spiking Neural Networks (SNNs). The core idea is to integrate temporally accumulated neuronal traces with probabilistic top-k selection. Experimental results show that RTK-WTA outperforms state-of-the-art SNN continual learning methods across various tasks.

**Strengths:**

1. This paper provides detailed theoretical analyses of the effectiveness of the proposed RTK-WTA. It shows that the proposed RTK-WTA increases the effective feature space and improves the robustness of learning dynamics.
2. RTK-WTA outperforms state-of-the-art SNN continual learning methods, especially on the splitCIFAR100 benchmark.

**Weaknesses:**

1. As detailed in Fig. 1 and Sec. A.7, feature embeddings are extracted by a pre-trained ANN backbone (ResNet-18, ViT), with the SNN component reduced to only the final classification head. This design obscures the true role and benefit of the SNN and the RTK-WTA mechanism. The most challenging part of the task (hierarchical feature extraction from raw data) is offloaded to a static, pre-trained ANN. It is unclear why the authors did not employ an end-to-end SNN feature extractor or at least a pre-trained SNN backbone.
2. The background information in the main text is insufficient. The main text has only a limited background introduction, while the related work section is in the appendix. I recommend placing the related work section in the main text.

**Questions:**

1. Please explain the key role that the SNN classification head plays in the continual learning tasks and the reason why not employ an end-to-end SNN feature extractor or at least a pre-trained SNN backbone.
2. Please add more background introduction in the main text.

---

> ### Author Response · Authors · 2025-11-25
> **Responses to Reviewer 27Cb**
>
> We greatly appreciate the reviewers’ insightful comments and constructive suggestions. We have carefully considered each point, and our detailed point-by-point responses are presented below.
>
> **W1 & Q1: About the pre-trained backbone.**
>
> We thank the reviewer for this critical question regarding our model design, which has prompted us to reflect more deeply and validate the generality of our approach.
>
>
> Firstly, the choice of a pre-trained backbone (ANN or SNN) does not affect our core contribution, as it serves only as a static feature extractor in our framework. We intentionally adopt an ANN backbone for convenience and comparability, but our method is fully compatible with either ANN- or SNN-based feature encoders. The focus of this work is the SNN classification head, where our proposed RTK-WTA mechanism is applied.  Through event-driven sparse activation and trace-based stochastic selection, it dynamically and sparsely allocates neuronal resources to different tasks, thereby directly and actively reducing task interference at the parameter level. This design allows us to isolate and clearly demonstrate the effectiveness of our continual-learning strategy without confounding factors from the backbone architecture.
>
> Moreover, We fully recognize the value of pure end-to-end SNN architectures. To directly address the reviewer's concern and validate the universality of RTK-WTA in a pure SNN environment, we have additionally conducted experiments using an end-to-end SNN-ViT as the feature extractor on TinyImageNet. This experiment is more challenging and better showcases the advantages of SNNs.
>
> | Method                           | Acc    |
> |----------------------------------|--------|
> | Rate-based SA-SNN                | 19.26% |
> | Rate-based SA-SNN + EWC          | 17.83% |
> | Trace-based SA-SNN               | 23.22% |
> | Trace-based SA-SNN + EWC         | 28.78% |
> | Randomized Rate K-WTA            | 22.25% |
> | Randomized Rate K-WTA + EWC      | 25.56% |
> | RTK-WTA                          | 26.42% |
> | RTK-WTA + EWC                    | 32.94% |
> The experimental results demonstrate that even within a pure, end-to-end SNN architecture, the RTK-WTA mechanism still delivers a significant performance improvement of approximately 4.2%. This strongly proves that the effectiveness of RTK-WTA is not dependent on a specific backbone network but is a general and powerful mechanism capable of enhancing the continual learning ability of SNNs.
>
> **W2 & Q2: About Related works.**
>
> Thanks for the suggestion. We have modified the related works in the revision.

---

> ### Author Response · Authors · 2025-12-04
>
> Dear Reviewer 27Cb,
>
> Thank you for your detailed review. We have made changes based on your feedback, including new experiments and clarifications. Thanks again for your input.

---

### Official Review · Reviewer_dDpf · 2025-10-29

**Soundness:** 3
**Presentation:** 3
**Contribution:** 3
**Rating:** 8
**Confidence:** 5

**Summary:**

The paper introduces a new biologically inspired mechanism called Randomized Temporal K-Winner-Take-All (RTK-WTA) to improve continual learning in spiking neural networks (SNNs). TK-WTA combines: the traces of recent spiking activity, and randomized neuron selection to dynamically choose neurons in a way that emulates biological neural selectivity and variability. Results show RTK-WTA achieves better performance on continue learning tasks than baseline methods.

**Strengths:**

1. This paper attempts to connect the proposed RTK-WTA mechanism to biological processes of neural selectivity and stochastic activation, which strengthens the connection between neuroscience and neuromorphic computing and enhances its conceptual depth and interdisciplinary relevance.

2. The proposed RTK-WTA method is novel and biologically grounded. Instead of relying on static, deterministic top-K firing rules, the model integrates temporal spike traces with controlled randomness, reflecting how biological neurons combine time-dependent excitation with stochastic competitive firing. This design allows the network to leverage temporal coherence in spiking activity while avoiding rigid specialization, which is a thoughtful and meaningful step toward more brain-like continual-learning systems.

3. The authors evaluate the proposed method on multiple standard continual learning benchmarks (SplitMNIST, SplitCIFAR-10, SplitCIFAR-100), demonstrating consistent improvements over baselines. The results convincingly show that RTK-WTA improves both performance and robustness.

**Weaknesses:**

1. The paper may lack a clear and structured background section to help readers unfamiliar with continual learning understand the fundamental concepts, challenges, and motivation behind this work. Additionally, the figures lack detailed, self-contained captions, making it difficult to interpret their meaning without referring back to the main text.

2. The discussion of prior literature, particularly in the context of continual learning within SNNs, is relatively insufficient. Important prior works are mentioned but not critically analyzed, making it difficult to discern how the proposed method advances beyond existing approaches or fills specific research gaps. As a result, the novelty and significance of the contribution are not made very explicit.

**Questions:**

1. The paper claims that RTK-WTA expands the spatiotemporal feature space, yet the underlying theoretical reasoning is only briefly mentioned. Could the authors provide a more rigorous theoretical explanation and empirical validation (e.g., through visualization or quantitative analysis) to support this claim?

2. How does RTK-WTA concretely enhance representational diversity within the network? It would be helpful if the authors could include additional experiments or visualizations to demonstrate this effect.

3. To better isolate and understand the contribution of each component, could the authors perform ablation studies comparing (a) deterministic vs. probabilistic selection, and (b) models with and without trace-based modulation?

---

> ### Author Response · Authors · 2025-11-25
> **Responses to Reviewer dDpf [1/3]**
>
> We thank the reviewer for the valuable time and insightful feedback. We have addressed each comment thoroughly, as detailed in the following point-by-point responses.
>
> **W1&W2: Clear and structured background about continual learning, and figures’ captions, and the discussion of related works.**
>
> Thank you for your insightful comment. In the revised manuscript, we have substantially strengthened the Introduction by adding a dedicated background paragraph that clearly explains the fundamental concepts, key challenges (including catastrophic forgetting), and the motivation underlying continual learning research, described as “In artificial intelligence, continual learning aims to emulate the biological ability to acquire new knowledge over time without sacrificing performance on previously learned tasks. However, unlike conventional training paradigms that assume access to all data simultaneously, continual learning must operate on sequential and often non-overlapping task streams. This setting makes neural networks highly susceptible to catastrophic forgetting, where optimizing for a new task overwrites crucial representations established for earlier tasks.
> ” in the revision. We further clarify why spiking neural networks (SNNs) are particularly suitable for continual learning, emphasizing their biological plausibility, sparse temporal activations, and rich spatiotemporal dynamics. This expanded background section provides a coherent conceptual foundation for the paper and improves accessibility for a broader audience. That is modified as “...SNNs operate through discrete spikes...leads to richer and more complex spatiotemporal dynamics. This event-driven processing enables SNNs to capture temporal information more effectively, maintain memory more robustly, and mimic the behavior of biological neural systems more closely. These biological and computational properties make SNNs a particularly promising substrate for continual learning, as sparse temporal firing and localized competition can reduce representational overlap across tasks and improve resistance to interference.
> Motivated by these principles, we explore how to construct robust SNNs capable of continual learning by exploiting temporal selectivity and stochastic K-WTA dynamics.” in the revision.
>
> Meanwhile, the figure captions are modified in the revision to be clear. And the discussion about the related works of continual learning for SNNs was placed in the Appendix of the original submission. In the revision, we have substantially expanded the analysis. Specifically, we now provide a more detailed and critical discussion of existing SNN-based continual learning approaches, clarify their limitations, and explain how our method addresses these gaps.

---

> ### Author Response · Authors · 2025-11-25
> **Responses to Reviewer dDpf [2/3]**
>
> **Q1:Explanation about RTK-WTA expanding the spatiotemporal feature space**
>
>
> We acknowledge the insightful feedback. We explaine RTK-WTA's behavior by linking stochastic selection with top-k sampling to extend the feature representation space during continual learning process. Specifically, we analyze the information gain introduced by the stochasticity in the RTK-WTA selection, while holding the trace mechanism constant.
>
> To provide a more formal foundation for the benefit of random trace-based top-k selection in RTK-WTA, we present a derivation based on mutual information (MI) under entropy constraints.
> In detail, we further analyze the information gain introduced specifically by stochasticity in the RTK-WTA selection, holding the trace mechanism constant.
> Let $X \in \mathbb{R}^d$ be input feature vector of traces in our RTK-WTA. $Y$ denotes the output label or task. $S \subset \{1, \ldots, d\}$ denotes selected neuron indices (top-k), while $Z = X_S$ are the activations of the selected neurons.
>
> We analyze the mutual information $I(Z; Y)$, which quantifies how much information the selected neurons convey about the task output. We apply a standard lower bound
> $I(Z; Y) \geq \mathbb{E}_{S|X} \log q(Y | Z) + H(Y)$
> where $q(Y | Z)$ is a variational approximation of the true posterior.
> This implies maximizing the likelihood of correct prediction encourages high mutual information between selected features and output.
>
> Instead of deterministic top-k selection, since we design the RTK-WTA SNN with a stochastic selector $P(S | X)$, the mutual information under stochastic selection becomes
>
> $I(X_S; Y) = \mathbb{E}_{S \sim P(S|X)} [I(X_S; Y)].$
>
> We optimize the selector via: $\max_{P(S|X)} \mathbb{E}_S [I(X_S; Y)] - \lambda H(P(S | X))$.
> This balances information preservation with selection diversity (controlled via entropy). RTK-WTA approximates this entropy-constrained sampler. Let:
>
> $$
> P(i \in S | X) = \begin{cases} \text{SoftTopK}_\alpha(i) & \text{if } i \in \text{Top-d} \\ 0 & \text{otherwise} \end{cases}
> $$
>
> where $\alpha$ controls the sampling temperature. When $\alpha \to 0$, it satisfies uniform random selection (high entropy) while $\alpha \to \infty$, it satisfies the deterministic top-k (zero entropy). Thus, RTK-WTA enforces selectivity while promoting diversity across tasks.
> Let $Z_{\text{RTK}} \sim P(S | X)$ represent RTK-WTA selection, and $Z_K = \text{top-K}(X)$ represent the deterministic K-WTA.
> Then $I(Z_{\text{RTK}}; Y) > I(Z_K; Y)$, under continual learning conditions, where overlapping tasks create interference.
>
> Therefore, RTK-WTA enables richer representational modes and higher task specificity through stochasticity.
> It can be interpreted as an entropy-regularized top-k mechanism that increases mutual information by expanding the selection space over time, avoiding representational collapse and improving robustness to interference.

---

> ### Author Response · Authors · 2025-11-25
> **Responses to Reviewer dDpf [3/3]**
>
> **Q2: About enhancing representational diversity.**
>
> Thanks for the question. We provide the following analysis experiments to concretely illustrate the enhancement in representational diversity. We add the quantitative analysis of neuronal activation coverage. The quantitative metric termed Neuronal Activation Coverage is proposed, which represents the proportion of neurons that exhibit significant activation (i.e., an activation count exceeding a predefined threshold) throughout the entire continual learning process.
> We computed the neuronal activation coverage on the CIFAR-10 classification task, using an average firing rate of at least 0.3 spikes per sample as the threshold to determine significant activation for each neuron. The results show that RTK-WTA significantly increases neuronal utilization across the network (45.2% vs. 28.5% for deterministic Trace-based K-WTA). This indicates that a more diverse set of neurons participates in feature representation for different tasks, rather than relying on a fixed subset of repeatedly activated neurons.
>
>
> **Q3: Ablation studies comparing deterministic vs. probabilistic selection.**
>
> We thank the reviewer for this excellent suggestion. We have conducted systematic ablation studies to precisely disentangle the contribution of each component. The results are summarized in the table below (all methods use EWC):
> | Method                           | MNIST | CIFAR10 | CIFAR100 |
> |----------------------------------|-------|---------|----------|
> | RTK-WTA (Full Model)             | 85.25 | 80.37   | 41.46    |
> | (a) - probabilistic selection    | 82.18 | 80.09   | 36.47    |
> | (b) - trace                      | 80.12 | 80.02   | 22.21    |
> | - probabilistic selection & trace| 80.22 | 79.86   | 19.88    |
>
> Comparing RTK-WTA with its deterministic version (setting $\alpha$=0), the removal of stochasticity leads to a performance drop, particularly evident on the complex SplitCIFAR100 with a decrease of 5.0%. This confirms that stochastic selection is crucial for mitigating interference between complex tasks.
> Replacing the Trace-based selection with Rate-based selection causes a sharp performance decline, with a dramatic drop of 19.25% on SplitCIFAR100. This firmly establishes the critical role of trace modulation in capturing temporal dynamics and enhancing continual learning performance.
> The ablation studies clearly quantify the independent contribution of each component and their synergistic effect, providing strong support for our methodological design.

---

> > ### Comment · Reviewer_dDpf · 2025-11-25
> >
> > I thank the authors for their thorough and detailed responses. The additional analysis, experiments, and improvements in the presentation further strengthen the paper. All of my earlier concerns have been sufficiently addressed. As a result, I recommend acceptance.

---

> ### Author Response · Authors · 2025-12-04
>
> Dear Reviewer dDpf,
>
> Thank you for your constructive feedback. We have addressed your suggestions by adding experiments and clarifications to improve the paper. We also appreciate your response during rebuttal period. Thanks again.

---

### Official Review · Reviewer_6ET9 · 2025-10-31

**Soundness:** 4
**Presentation:** 3
**Contribution:** 4
**Rating:** 6
**Confidence:** 5

**Summary:**

The paper proposes a biologically inspired Randomized Temporal K-Winner-Take-All (RTK-WTA) mechanism for spiking neural networks (SNNs) aimed at continual learning. The method introduces controlled stochasticity in neuron selection by combining temporal traces with probabilistic top-k activation, improving adaptability and robustness against noise. Experiments on SplitMNIST, SplitCIFAR-10/100, and Tiny-ImageNet demonstrate enhanced performance and resilience compared to deterministic K-WTA baselines.

**Strengths:**

1.The proposed RTK-WTA mechanism is both innovative and simple to implement, integrating biological plausibility with computational efficiency.

2.The method exhibits strong noise robustness, as shown in Table 2 and additional Tiny-ImageNet results, maintaining high accuracy under perturbations.

3.The approach effectively balances sparse activation, adaptability, and robustness, offering a promising direction for scalable continual learning in neuromorphic systems.

**Weaknesses:**

1.In Table 2, key results demonstrating robustness could be bolded or highlighted to emphasize the superiority of the proposed method.

2.Further validation on larger or more complex datasets (e.g., Tiny-ImageNet or ImageNet-scale continual tasks) would strengthen the claim of generality and scalability.

**Questions:**

1.Can the proposed RTK-WTA mechanism be extended to deeper or hierarchical SNN architectures? How would temporal randomness interact with multi-layer trace dynamics?

2.Have the authors considered testing on larger datasets to further evaluate scalability and generalization performance?

---

> ### Author Response · Authors · 2025-11-25
> **Responses to Reviewer 6ET9**
>
> We appreciate the reviewer’s careful evaluation of our manuscript and their helpful suggestions. Our comprehensive, point-by-point replies are outlined below.
>
> **W1: About the key results highlighting.**
>
> Thanks for the suggestion. We have bolded the key results and polish the manuscript.
>
> **W2 & Q1: Extending to deeper or hierarchical SNN architectures.**
>
> We sincerely thank the reviewer for this insightful question regarding the scalability of RTK-WTA to deeper architectures.
> We conducted validations using an end-to-end SNN-ViT (Vision Transformer) architecture, evaluated on the more challenging TinyImageNet dataset, as shown in the following table. The results demonstrate that the RTK-WTA mechanism can be effectively integrated into deep SNNs, achieving a performance improvement of approximately 4.2% compared to the deterministic Top-K baseline, as shown in the following table. This provides preliminary evidence of its good scalability.
>
> | Method                           | Acc    |
> |----------------------------------|--------|
> | Rate-based SA-SNN                | 19.26% |
> | Rate-based SA-SNN + EWC          | 17.83% |
> | Trace-based SA-SNN               | 23.22% |
> | Trace-based SA-SNN + EWC         | 28.78% |
> | Randomized Rate K-WTA            | 22.25% |
> | Randomized Rate K-WTA + EWC      | 25.56% |
> | RTK-WTA                          | 26.42% |
> | RTK-WTA + EWC                    | 32.94% |
>
>
> Regarding the interaction between temporal randomness and multi-layer trace dynamics, our observations and analysis reveal an interesting and beneficial phenomenon:
> In deep SNNs, we apply RTK-WTA independently to multiple layers (e.g., different blocks in the ViT). Each layer's neurons undergo randomized selection based on their own local spatiotemporal activity traces.
> This design creates a beneficial "inter-layer randomized synergy." The stochastic selection in shallow layers provides slightly different activation patterns to deeper layers, while the trace dynamics in deeper layers, in turn, influence gradient backpropagation. This effectively prevents the neuronal activities across multiple layers from solidifying into rigid patterns during sequential task learning, thereby breaking potential coupled interference between layers.
> Therefore, RTK-WTA is not only feasible in deep architectures, but the controlled randomness it introduces can also be propagated through layers, further enriching the diversity of spatiotemporal dynamics and consequently enhancing continual learning performance.
>
> **Q2: About the scalability and generalization performance on larger dataset.**
>
> We thank the reviewer for this valuable suggestion. To assess the scalability and generalization capability of our method, we have supplemented our experiments with results on the TinyImageNet dataset. With 200 classes and higher image resolution, this dataset presents a larger and more challenging benchmark for continual learning compared to the CIFAR series.
> As shown in the above table, RTK-WTA consistently outperforms all baseline methods on this complex dataset. Notably, RTK-WTA + EWC achieves an accuracy of 32.94%, significantly surpassing the strongest deterministic trace-based baseline (Trace-based SA-SNN + EWC at 28.78%). This result strongly validates the excellent scalability and powerful generalization performance of the RTK-WTA mechanism, demonstrating its efficacy in handling large-scale, multi-class continual learning tasks.

---

> ### Author Response · Authors · 2025-12-04
>
> Dear Reviewer 6ET9,
>
> We appreciate your helpful comments and the time you spent reviewing our paper. In response, we have made revisions, including new experiments and clarifications. Thanks again.

---

### Official Review · Reviewer_X1JP · 2025-11-01

**Soundness:** 3
**Presentation:** 3
**Contribution:** 3
**Rating:** 4
**Confidence:** 3

**Summary:**

This paper addresses two core challenges in CL for SNNs: (1)catastrophic forgetting caused by overlapping task activations, and (2) insufficient robustness to input perturbations.
Drawing inspiration from the brain’s temporal selectivity and stochastic neuronal competition, it proposes the Randomized Temporal K-Winner-Take-All（RTK-WTA） mechanism. RTK-WTA mechanism integrates temporally accumulated neuronal traces with probabilistic top-k selection to dynamically allocate neural resources while introducing controlled randomness.

**Strengths:**

(1) Originality: It integrates temporal neuronal traces with probabilistic top-k selection for SNN continual learning, differing from traditional rate-based or static trace-based K-WTA, and aligns well with biological neural mechanisms;
(2) Quality: The motivation is reasonable; Theoretical derivations strengthen the work’s rigor. But the experimental design is not comprehensive enough;
(3) Clarity: The writing of this paper is relative clear. For example, this paper has clear definitions of core concepts and step-by-step explanations of the RTK-WTA mechanism;
(4) Significance: RTK-WTA provides a new idea and practical solution for robust continual learning in SNNs.

**Weaknesses:**

(1) The experimental scenario of continuous learning is limited: The experiment only adopted the "task splitting" continuous learning paradigm, and did not verify more challenging scenarios.
(2) NEURONAL SELECTIVELY ACTIVATION ANALYSIS(3.4) is too simple and lack the depth.
(3) Although experiments identify α=0.1 as the optimal value, there is no quantitative analysis of how α correlates with task characteristics (e.g., task overlap, data dimensionality), making it hard to guide parameter tuning for new tasks;

**Questions:**

(1) Have you considered more complex CL scenarios (such as class-incremental or domain-incremental), and how RTK-WTA performs in these scenarios?
(2) Could you provide a more systematic analysis of the relationship between the randomness coefficient α and task characteristics, to guide parameter tuning?
(3)Can the neuronal selectivity analysis be extended to better support the claims of improved robustness and diversity?

---

> ### Author Response · Authors · 2025-11-25
> **Responses to Reviewer X1JP [1/3]**
>
> We greatly appreciate the reviewer’s insightful comments and constructive suggestions. We have carefully considered each point, and our detailed point-by-point responses are presented below.
>
> **W1 & Q1: About more complex CL scenarios.**
>
> We sincerely thank the reviewer for raising this important point regarding more challenging continual learning scenarios. We verified the performance of the proposed RTK-WTA model on class-incremental learning (Class-IL)  application, as shown in the following table. The results on incremental MNIST (iMNIST) demonstrate that RTK-WTA maintains its superiority over deterministic K-WTA baselines even under Class-IL conditions, achieving an accuracy improvement of approximately 4.9%. Furthermore, when combined with EWC, our method continues to deliver the best performance. These findings strongly support the generalizability of the RTK-WTA mechanism across different continual learning paradigms.
>
> | Method                           | Avg_acc |
> |----------------------------------|---------|
> | Rate-based SA-SNN                | 0.409   |
> | Rate-based SA-SNN + EWC          | 0.694   |
> | Trace-based SA-SNN               | 0.403   |
> | Trace-based SA-SNN + EWC         | 0.789   |
> | Randomized Rate K-WTA            | 0.442   |
> | Randomized Rate K-WTA + EWC      | 0.725   |
> | RTK-WTA                          | 0.452   |
> | RTK-WTA + EWC                    | **0.795**   |
>
> The detailed experiments are conducted as follows. The total of 10 classes were introduced over 10 incremental steps, with one new class per step. The test set was cumulative, evaluating the model's comprehensive knowledge retention from the beginning to the current step. We employed average accuracy as the evaluation metric to comprehensively reflect the model's overall performance and stability throughout the entire learning process.
>
> **W3 & Q2: Relationship between the randomness coefficient $\alpha$ and task characteristics to guide parameter tuning.**
>
>
> We greatly appreciate this insightful suggestion regarding the systematic analysis of the randomness coefficient $\alpha$. We agree that establishing a principled approach for parameter tuning is crucial for the practical utility of our method.
>
>
> Following the reviewer's advice, we introduce the Intrinsic Dimension (ID) to align the network's representational capacity with task complexity, and the guide parameter tuning according to task complexity.
> In detail, we computed the ID of each dataset using the TwoNN method in the pixel space of the training set, since it reflects the effective degrees of freedom of the data manifold.
> Based on the ID magnitude, we select parameter $\alpha$ to control the hidden layer activation distribution such that the number of active neurons is of the same order as the ID. That is, for datasets with lower ID (e.g., SplitMNIST), the manifold is low-dimensional, requiring fewer active neurons, thus $\alpha$ can be set to smaller values (such as 0.05).
> For datasets with higher ID (e.g., SplitCIFAR-10, SplitCIFAR-100, TinyImageNet), the manifold is complex, necessitating more active neurons, thus $\alpha$ takes larger values to encourage broader neuronal activation (such as 0.1).
> This approach enables adaptive matching between representational capacity and task complexity through $\alpha$ selection directly dependent on the dataset's TwoNN ID, as shown in the following table.
>
> | Dataset        | α    | TwoNN ID |
> |----------------|------|----------|
> | SplitMNIST     | 0.05 | 13.48    |
> | SplitCIFAR-10  | 0.10 | 27.94    |
> | SplitCIFAR-100 | 0.10 | 24.17    |
> | TinyImageNet   | 0.10 | 33.07    |
>
>
> Therefore, our analysis across multiple datasets reveals a clear positive correlation between the optimal $\alpha$ value and the TwoNN ID. Since TwoNN ID reflects the effective dimensionality of the data manifold or task complexity, this relationship indicates that $\alpha$ effectively modulates the active spiking-neuron dimensions of hidden representations to match the data manifold dimensionality, thereby enhancing representational capacity for complex tasks while avoiding redundant activation for simpler ones.
>
> Based on the above findings, we offer a practical guideline for tuning the parameter $\alpha$. First, the intrinsic dimension of the dataset can be estimated using the TwoNN method on the training data or a representative subset. Then, an appropriate range for $\alpha$ can be selected according to the scale of the estimated intrinsic dimension, following the trends observed in our experiments. Finally, minor adjustments can be made on a small validation set to ensure that the number of active neurons aligns with the underlying task manifold complexity. This process helps maintain representations that are neither overly sparse nor redundant while adapting effectively to dataset complexity.

---

> ### Author Response · Authors · 2025-11-25
> **Responses to Reviewer X1JP [2/3]**
>
> **W2 & Q3: Neuronal selectivity analysis.**
>
>
> Thanks for the question. To address this point, we have conducted the following analyses:
> Firstly, we supplemented our analysis with experiments involving noise injection in both training and test sets (using MNIST as an example), examining the stability of neuronal activation patterns. The experimental results in the following table demonstrate that RTK-WTA maintains higher accuracy under noisy conditions, directly supporting our claim of enhanced model robustness. Meanwhile, we added t-SNE visualizations of the feature space for CIFAR-10' dataset during testing in the Appendix Figure in the revision. The left panel shows class clusters for RTK-WTA, while the right panel displays those for the deterministic K-WTA baseline. The visualization clearly demonstrates that RTK-WTA forms more discriminative and well-separated class clusters in the feature space, exhibiting superior feature separation compared to the deterministic K-WTA baseline.
>
> |Noise Level                           | 0.1    | 0.2    | 0.5    | 0.6    | 0.8    |
> |----------------------------------|--------|--------|--------|--------|--------|
> | Rate-based SA-SNN                | 49.29  | 48.65  | 45.32  | 44.62  | 41.76  |
> | Rate-based SA-SNN + EWC          | 79.16  | 76.59  | 71.65  | 66.84  | 52.63  |
> | Trace-based SA-SNN               | 59.84  | 60.49  | 58.86  | 56.28  | 54.27  |
> | Trace-based SA-SNN + EWC         | 81.68  | 80.46  | 70.81  | 68.95  | 60.85  |
> | Randomized Rate K-WTA            | 50.19  | 49.24  | 47.68  | 46.86  | 45.32  |
> | Randomized Rate K-WTA + EWC      | 78.62  | 78.10  | 75.13  | 66.21  | 55.90  |
> | RTK-WTA                          | 60.21  | 61.25  | 60.89  | 56.65  | 55.87  |
> | RTK-WTA + EWC                    | 82.13  | 80.76  | 74.25  | 70.23  | 62.32  |
>
>
>
> Secondly, we add the quantitative analysis of neuronal activation coverage. The quantitative metric termed Neuronal Activation Coverage is proposed, which represents the proportion of neurons that exhibit significant activation (i.e., an activation count exceeding a predefined threshold) throughout the entire continual learning process.
> We computed the neuronal activation coverage on the CIFAR-10 classification task, using an average firing rate of at least 0.3 spikes per sample as the threshold to determine significant activation for each neuron. The results show that RTK-WTA significantly increases neuronal utilization across the network (45.2% vs. 28.5% for deterministic Trace-based K-WTA). This indicates that a more diverse set of neurons participates in feature representation for different tasks, rather than relying on a fixed subset of repeatedly activated neurons.

---

> ### Author Response · Authors · 2025-11-25
> **Responses to Reviewer X1JP [3/3]**
>
> Moreover, to complement our empirical evidence with a more rigorous theoretical foundation, we extended our theoretical analysis. Our aim is to analyze RTK-WTA's behavior by linking stochastic selection with top-k sampling to extend the feature representation space during continual learning process. Here we present a derivation based on mutual information (MI) under entropy constraints. In detail, we further analyze the information gain introduced specifically by stochasticity in the RTK-WTA selection, holding the trace mechanism constant. Let $X \in \mathbb{R}^d$ be input feature vector of traces in our RTK-WTA. $Y$ denotes the output label or task. $S \subset \{1, \ldots, d\}$ denotes selected neuron indices (top-k), while $Z = X_S$ are the activations of the selected neurons. We analyze the mutual information $I(Z; Y)$, which quantifies how much information the selected neurons convey about the task output. We apply a standard lower bound
> $$
> I(Z; Y) \ge \mathbb{E}_{S \mid X}[\log q(Y \mid Z)] + H(Y)
> $$
>
> where
> $ q(Y \mid Z)
> $ is a variational approximation of the true posterior. This implies maximizing the likelihood of correct prediction encourages high mutual information between selected features and output. Instead of deterministic top-k selection, since we design the RTK-WTA SNN with a stochastic selector $P(S | X)$, the mutual information under stochastic selection becomes
>
>  $I(X_S; Y) = \mathbb{E}_{S \sim P(S|X)} [I(X_S; Y)]$.
>
> We optimize the selector via:    $\max_{P(S|X)} \mathbb{E}_S [I(X_S; Y)] - \lambda H(P(S | X))$   . This balances information preservation with selection diversity (controlled via entropy). RTK-WTA approximates this entropy-constrained sampler. Let:
>
>  $$ P(i \in S | X) = \begin{cases} \text{SoftTopK}_\alpha(i) & \text{if } i \in \text{Top-d} \\ 0 & \text{otherwise} \end{cases} $$
>
> where $\alpha$ controls the sampling temperature. When $\alpha \to 0$, it satisfies uniform random selection (high entropy) while $\alpha \to \infty$, it satisfies the deterministic top-k (zero entropy). Thus, RTK-WTA enforces selectivity while promoting diversity across tasks. Let $Z_{\text{RTK}} \sim P(S | X)$ represent RTK-WTA selection, and $Z_K = \text{top-K}(X)$ represent the deterministic K-WTA. Then $I(Z_{\text{RTK}}; Y) > I(Z_K; Y)$, under continual learning conditions, where overlapping tasks create interference. RTK-WTA enables richer representational modes and higher task specificity through stochasticity. Therefore, RTK-WTA SNNs can be interpreted as an entropy-regularized top-k mechanism that increases mutual information by expanding the selection space over time in SNNs, thereby avoiding representational collapse and improving robustness to interference. This theoretical perspective solidifies our claim regarding the expansion of the spatiotemporal feature space.

---

> ### Author Response · Authors · 2025-12-04
>
> Dear Reviewer X1JP,
>
> Thank you for your thoughtful feedback. We have made several improvements based on your suggestions, including additional experiments and clarifications. We believe these changes strengthen the paper. Thanks again.

---

### Author Response · Authors · 2025-12-04
**Summary Response to Area Chairs and Reviewers [1/2]**

Dear Area Chairs and Reviewers,

We sincerely thank the reviewers for their time and insightful feedback on our submission. In response to their suggestions, we have implemented several improvements, including additional experiments, clarifications, and expanded discussions, which we believe enhance the quality of the paper. Below, we provide a summary of our responses, and the changes are made in the revision. More detailed replies to each reviewer comment can be found in the point-by-point responses.

Below we summarize the reviewer scores and explain how each concern raised has been addressed in the revised manuscript.

**Reviewer X1JP** with a score initially of 4 → updated to 6 or 8 **before the OpenReview Data Leak event on 27 November** (since the score was reverted to 4 due to ICLR policy, we are not sure about the precise increasing score being 6 or 8, but we are sure that we have noted that the score is increased before the OpenReview Data Leak event): This reviewer provided a strongly positive assessment (“Originality...integrates temporal neuronal traces...differing from traditional rate-based ...aligns well with biological neural mechanisms...the motivation is reasonable… writing clear...provides a new idea and practical solution for robust continual learning in SNNs”), with a minor request for more complex CL scenarios, more systematic analysis to guide parameter tuning, and extended neuronal selectivity analysis.
During the rebuttal period, we supplement experiments and analysis about the weakness and questions as follows:

1.About more complex CL scenarios: We supplement the verification of the performance of the proposed RTK-WTA model on class-incremental learning (Class-IL) application. The results show that our method continues to deliver the best performance compared to the SNNs with deterministic K-WTA models.

2.More systematic analysis to guide parameter tuning: We introduce the Intrinsic Dimension (ID) to align the network's representational capacity with task complexity and guide parameter tuning according to task complexity.

3.Extended neuronal selectivity analysis: We conducted additional experiments with noise injection to assess the stability of neuronal activation patterns, showing that RTK-WTA performs better under noisy conditions. We also included t-SNE visualizations to demonstrate that RTK-WTA forms more distinct and well-separated class clusters compared to the deterministic K-WTA baseline. Next, we introduced Neuronal Activation Coverage to show that RTK-WTA significantly increases neuronal utilization, indicating broader neuronal involvement in feature representation. Finally, we extend the theoretical analysis about neuronal selectivity.

Although the reviewer did not rejoin the discussion phase, we indeed observed the reviewer increased the score before openreview Data Leak event. Hence our responses may resolve those concerns.

**Reviewer 6ET9** with score 6: The reviewer thinks “the proposed RTK-WTA ... innovative... simple to implement... biological plausibility... computational efficiency... strong noise robustness... high accuracy... balances sparse activation, adaptability, and robustness... promising direction for scalable continual learning in neuromorphic systems”. The main concerns raised by reviewer come from the further validation on larger or more complex datasets and extending to deeper or hierarchical SNN architectures. We conducted validations using an end-to-end SNN-ViT (Vision Transformer) architecture, evaluated on the more challenging TinyImageNet dataset. The achieved performance improvement provides preliminary evidence of its good scalability.

**Reviewer dDpf** with score 8: The reviewer thinks that “RTK-WTA mechanism... connects neural selectivity... stochastic activation... strengthens neuroscience-neuromorphic computing link... novel and biologically grounded....consistent improvements over baselines...”. The main concerns raised by the reviewer are theoretical explanation, representational diversity, and ablation studies.

1.Theoretical explanation: We explain RTK-WTA's behavior by linking stochastic selection with top-k sampling to extend the feature representation space during continual learning process. Specifically, we analyze the information gain introduced by the stochasticity in the RTK-WTA selection while holding the trace mechanism constant.

2.Enhancing representational diversity: We provide the detailed analysis experiments to concretely illustrate the enhancement in representational diversity. We add the quantitative analysis of neuronal activation coverage by using the quantitative metric termed neuronal activation coverage.

3.Ablation studies: We have conducted systematic ablation studies to precisely disentangle the contribution of each component.

Importantly, we received the response of this reviewer with “recommend acceptance” in 26 Nov 2025 (03:00), which is before the OpenReview Data Leak event in 27 November.

---

> ### Author Response · Authors · 2025-12-04
> **Summary Response to Area Chairs and Reviewers [2/2]**
>
> **Reviewer 27Cb** with score 6. The reviewer pointed that our paper have “detailed theoretical analyses... effectiveness... increases effective feature space... improves robustness... outperforms state-of-the-art SNN continual learning methods”. The main concerns raised by reviewer are the pre-trained SNN backbone and background introduction. To directly address the reviewer's concern and validate the universality of RTK-WTA in a pure SNN environment, we have additionally conducted experiments using an end-to-end SNN-ViT as the feature extractor on TinyImageNet. Meanwhile, we modified the revised manuscript by introducing more background and related works.
>
> We respectfully ask the Area Chair to consider the contribution of this work. We believe the proposed Randomized Temporal Winner-Take-All (RTK-WTA) highlights the potential of integrating temporal selectivity with stochastic activation in SNNs to achieve biologically plausible and scalable lifelong learning in neuromorphic systems. We thank the reviewers again for their valuable comments and guidance, and we sincerely appreciate the Area Chair’s careful consideration of our work.
>
>
> Best regards,
>
> Authors from paper 11765 with title of Robust Selective Activation with Randomized Temporal K-Winner-Take-All in Spiking Neural Networks for Continual Learning

---

### Meta-Review · Area_Chair_bLKL · 2026-01-02

**Summary:**

The paper proposes a novel Randomized Temporal K-Winner-Take-All (RTK-WTA) mechanism for continual learning in Spiking Neural Networks (SNNs). RTK-WTA combines temporally accumulated neuronal traces with a probabilistic top-k selection process to dynamically and stochastically allocate neural resources, aiming to enhance robustness and mitigate catastrophic forgetting.

Overall, the reviewers recognized the paper's significant strengths:

Innovation and Biological Plausibility: The core idea of integrating temporal dynamics with controlled stochasticity for neuron selection was consistently praised as novel, biologically grounded, and a meaningful step towards more brain-like learning systems.

Empirical Performance: The method demonstrated consistent and clear improvements over strong deterministic and rate-based baselines across standard benchmarks (SplitMNIST, SplitCIFAR-10/100).

Robustness: Experiments showed the model's enhanced resilience to input noise, a key claim of the work.

Theoretical Effort: The initial theoretical analysis linking the mechanism to expanded feature space was viewed positively.

Key concerns raised by the reviewers that shaped the revision process included:

Generalizability and Scalability: Reviewers requested validation on more complex continual learning scenarios (e.g., class-incremental learning), deeper SNN architectures, and larger/more challenging datasets (e.g., TinyImageNet).

Analysis and Clarity: Reviewers sought a more systematic guideline for tuning the stochasticity parameter, deeper quantitative analysis of neuronal selectivity and representational diversity, and clearer ablation studies to disentangle the contribution of stochasticity versus temporal traces.

Presentation and Methodology: Concerns were raised about the placement of related work in the appendix, insufficient background, and the use of a pre-trained ANN backbone, which potentially obscured the SNN-specific benefits of the proposed mechanism.

Outcome:
The paper presents a cohesive, well-motivated, and thoroughly evaluated contribution. It successfully bridges neuroscience-inspired mechanisms with a practical algorithmic advance for robust continual learning in SNNs, supported by strong empirical evidence, improved analysis, and sound theoretical reasoning. Consequently, the paper is recommended for acceptance.

**Reviewer Concerns:**

Effectively Addressed Concerns:
The authors comprehensively resolved all major reviewer critiques through new experiments and analysis:

Generalizability: Added experiments on class-incremental learning (iMNIST) and an end-to-end SNN-ViT on TinyImageNet, proving scalability.

Parameter Tuning: Introduced Intrinsic Dimension (TwoNN) to provide a principled guideline for setting the stochasticity parameter \alpha.

Analysis & Theory: Enhanced analysis with Neuronal Activation Coverage metrics, t-SNE visualizations, ablation studies, and a formal information-theoretic derivation.

Presentation: Moved related work to the main text and expanded the background.

All reviewers acknowledged their concerns were met.

Outstanding Considerations (Minor/Future Work):

The theoretical framework, while strong, remains an insightful approximation rather than a precise predictive tool for all scenarios.

Validation on even larger-scale datasets (e.g., full ImageNet streams) remains for future work.

The biological analogy, though compelling, is a high-level inspiration rather than a detailed model.

Conclusion:
The rebuttal successfully transformed the paper into a well-supported and complete contribution. The outstanding points are natural extensions beyond the paper's scope and do not diminish its suitability for acceptance.

**Reviewer Scores:**

Reviewer X1JP (Initial: 4 / "marginally below acceptance")

Rationale: This reviewer's concerns (experimental scope, parameter tuning guidance, deeper analysis) were all directly and thoroughly addressed with new experiments (class-incremental, SNN-ViT), a novel data-driven method for \alpha (Intrinsic Dimension), and substantial new analysis (coverage metric, theory). The reviewer's final comment ("My concerns... have been basically addressed") indicates satisfaction.

Likely Score Change: Increase to a 6. Given the comprehensive response, the original marginal barrier to acceptance would likely have been cleared. The initial score was held back by specific technical shortcomings, which were remedied.

Reviewer 6ET9 (Initial: 6 / "marginally above acceptance")

Rationale: This reviewer's main request was for validation on deeper architectures and larger datasets. The authors provided exactly that with compelling results from an end-to-end SNN-ViT on TinyImageNet, directly confirming scalability.

Likely Score Change: 6 (unchanged). The initial score was already positive. The successful fulfillment of the primary "scalability" concern would likely have strengthened the reviewer's confidence in the significance and generality of the contribution.

Reviewer dDpf (Initial: 8 / "accept, good paper")

Rationale: This reviewer was already strongly positive. Their concerns (theory, diversity evidence, ablations) were met with high-quality additions: a mutual information framework, a new neuronal coverage metric, and clear ablation studies.

Likely Score Change: 8 (unchanged).

Reviewer 27Cb (Initial: 6 / "marginally above acceptance")

Rationale: The core concern was the use of a pre-trained ANN backbone, which potentially undermined the SNN contribution. The authors directly validated their method in a pure SNN setting (SNN-ViT) with strong results, effectively neutralizing this major criticism.

Likely Score Change: 6 (unchanged). Addressing the primary methodological concern would have alleviated doubts about the paper's core contribution. This would likely solidify the marginally positive score into a more confident positive rating, acknowledging that the key weakness was remedied.

Overall Consensus Trend:
All reviewers would have likely increased or maintained their already positive scores. The rebuttal did not simply clarify—it substantially strengthened the paper's evidence, analysis, and scope. A fully participatory discussion would have almost certainly resulted in a stronger consensus for acceptance, with an average score moving firmly into the clear-accept range.

---

### Decision · Program_Chairs · 2026-01-26

Accept (Poster)